# Growth, Proliferation and Metastasis of Prostate Cancer Cells Is Blocked by Low-Dose Curcumin in Combination with Light Irradiation

**DOI:** 10.3390/ijms22189966

**Published:** 2021-09-15

**Authors:** Jochen Rutz, Aicha Benchellal, Wajdi Kassabra, Sebastian Maxeiner, August Bernd, Stefan Kippenberger, Nadja Zöller, Felix K.-H. Chun, Eva Juengel, Roman A. Blaheta

**Affiliations:** 1Department of Urology, Goethe-University, 60590 Frankfurt am Main, Germany; Jochen.Rutz@kgu.de (J.R.); Aicha.Benchellal@hotmail.de (A.B.); w.kassabra@gmail.com (W.K.); Sebastian.Maxeiner@kgu.de (S.M.); Felix.Chun@kgu.de (F.K.-H.C.); eva.Juengel@unimedizin-mainz.de (E.J.); 2Department of Dermatology, Venereology, and Allergology, Goethe-University, 60590 Frankfurt am Main, Germany; bernd@em.uni-frankfurt.de (A.B.); kippenberger@em.uni-frankfurt.de (S.K.); nadjazoeller@netscape.net (N.Z.)

**Keywords:** curcumin, light irradiation, prostate cancer, growth, proliferation, metastasis

## Abstract

Although anti-cancer properties of the natural compound curcumin have been reported, low absorption and rapid metabolisation limit clinical use. The present study investigated whether irradiation with visible light may enhance the inhibitory effects of low-dosed curcumin on prostate cancer cell growth, proliferation, and metastasis in vitro. DU145 and PC3 cells were incubated with low-dosed curcumin (0.1–0.4 µg/mL) and subsequently irradiated with 1.65 J/cm^2^ visible light for 5 min. Controls remained untreated and/or non-irradiated. Cell growth, proliferation, apoptosis, adhesion, and chemotaxis were evaluated, as was cell cycle regulating protein expression (CDK, Cyclins), and integrins of the α- and β-family. Curcumin or light alone did not cause any significant effects on tumor growth, proliferation, or metastasis. However, curcumin combined with light irradiation significantly suppressed tumor growth, adhesion, and migration. Phosphorylation of CDK1 decreased and expression of the counter-receptors cyclin A and B was diminished. Integrin α and β subtypes were also reduced, compared to controls. Irradiation distinctly enhances the anti-tumor potential of curcumin in vitro and may hold promise in treating prostate cancer.

## 1. Introduction

Prostate cancer is the most common cancer in men, with 1,414,259 new cases and 375,304 recorded deaths worldwide in 2020 [1]. Standard treatment options depend on the tumor stage and include surgery, androgen deprivation therapy, radio-, chemo-, or immuno-therapy. Once the tumor has spread outside the prostate, the disease is difficult to treat with no curative option. Strong side effects and resistance development that inevitably occurs with conventional treatment lead many tumor patients to turn to complementary and alternative medicine (CAM). Up to 50% of cancer patients worldwide use CAM for all cancer [2], hoping to stop tumor growth, prolong survival, or even cure their disease [3].

During the last decade, reports pointing to the natural compound curcumin as a potential anti-cancer remedy have been published. Curcumin, a yellow-orange pigment extracted from the rhizome of the Curcuma longa plant, serves as a traditional food additive and spice ingredient. Preclinical studies have demonstrated that curcumin exerts anti-inflammatory, anti-oxidant, and anti-tumor activities [4]. In vitro, curcumin inhibits tumor cell proliferation and cell cycle progression and induces apoptosis [5,6,7]. There is also evidence that curcumin may counteract chemo- and radioresistance [8,9], making this compound interesting for clinical use. Unfortunately, the promising potential of curcumin has not been confirmed in clinical trials due to its low bioavailability. Poor water solubility, rapid metabolization, and low absorption do not allow ingested curcumin to develop its full potential [10]. Several methods have been undertaken to enhance its bioavailability, including the development of curcumin nanoparticles, phospholipid complexes, liposomal formulations, or structural analogs [10,11,12]. All of these approaches, however, have not led to satisfactory bioavailability. This is also true with respect to the influence of curcumin on prostate cancer. Although distinct effects have been noted in in vitro approaches, and although curcumin supplementation might have beneficial effects on some parameters related to prostate diseases [13], the overall clinical potential remains unsatisfactory. In fact, a recent interim analysis clearly showed that adding curcumin to metastatic castration-resistant prostate cancer patients was not effective in terms of progression-free survival [14].

Recently, Bernd presented evidence of increased antitumor properties of curcumin on human epidermal keratinocytes in vitro when irradiated with UVA or visible light [15]. The findings have been confirmed in a xenograft model, demonstrating tumor shrinkage when curcumin application was followed by irradiation [16]. The present investigation deals with whether this strategy might be of value in treating prostate cancer. To this end, the effect of curcumin on prostate cancer cell growth/proliferation and metastasis in vitro was analyzed in combination with light exposure. Cell cycling, cell cycle, and invasion related proteins were evaluated as well.

## 2. Results

### 2.1. Tumor Cell Growth

Low doses of curcumin of 0.1–0.4 µg/mL did not cause significant DU145 and PC3 growth reduction, compared to the untreated controls (Figure 1A). In contrast, curcumin application at all three dosages combined with exposure to visible light (5 min at 1.65 J/cm^2^) resulted in a significantly decreased growth rate after 48 and 72 h (Figure 1B). Curcumin at 0.2 µg/mL was chosen for all further experimentation since both PC3 and DU145 cells were strongly affected by this dose, but not so strongly inhibited that modified cell behavior may have been impossible to detect.

### 2.2. Clonogenic Growth

Figure 2, left, depicts morphology of tumor clones after 7 days of incubation. DU145 formed tight clone aggregates when grown in culture medium alone or treated with either visible light (5 min at 1.65 J/cm^2^) or curcumin (0.2 µg/mL). Combined treatment induced a distinct loss of cell–cell contact and destruction of clone formation. Similar clone behavior was seen with PC3 cells, although the clones were less densely packed. Figure 2, right, shows significant reduction of the cell clone number following tumor treatment with curcumin (0.2 µg/mL) plus irradiation (5 min visible light at 1.65 J/cm^2^). No effects were observed when DU145 or PC3 were treated with curcumin or light alone.

### 2.3. Tumor Cell Proliferation and Cell Cycling

Treatment with curcumin (0.2 µg/mL) and subsequent irradiation with visible light (5 min at 1.65 J/cm^2^) led to a significant decrease in proliferation of both DU145 and PC3 cells. No significant effects, compared to untreated controls, were apparent when cells were treated with light or curcumin alone (Figure 3A). In analyzing cell cycle progression, low-dosed curcumin (0.2 µg/mL) alone as well as light alone did not alter G0/G1-, S-, and G2/M-phases, compared to the controls, whereas curcumin combined with light did (Figure 3B). The number of DU145 cells undergoing the G0/G1-phase decreased, whereas the number of G2/M-phase cells was elevated. An increase in G2/M-phase cells was also seen in PC3 cultures which was accompanied by a decrease in S-phase cells.

### 2.4. Apoptosis

Figure 4 indicates the percentage of vital (lower left quadrant) and necrotic cells (upper left quadrant), and cells in early (lower right quadrant) and late apoptosis (upper right quadrant). Visible light irradiation (5 min at 1.65 J/cm^2^) or curcumin (0.2 µg/mL) did not result in significant alterations of apoptosis, related to the controls. However, the percentage of apoptotic DU145 and PC3 cells considerably increased when treated with light and curcumin (Figure 4A,B). PARP cleavage became evident in presence of light and curcumin but not in presence of light or curcumin alone (Figure 4C).

### 2.5. Cell Cycle Regulating Protein Expression

Cell cycle regulating proteins were not altered by curcumin (0.2 µg/mL) or visible light alone (5 min at 1.65 J/cm^2^), excepting CDK2, which was diminished in DU145 and PC3 cells and Rictor, which was suppressed in PC3 cells with all treatment modalities (Figure 5 and Appendix A). Exposing the tumor cells to visible light and curcumin was associated with distinct modifications of the protein expression level. CDK1 (both total and phosphorylated) along with cyclin A and B were suppressed. pCDK2 was diminished as well. Akt and pAkt were also downregulated by curcumin plus light. The same was seen with pRictor and pRaptor (DU145 > PC3), whereas no specific effects were seen with Rictor and Raptor (total).

### 2.6. Adhesion and Migration Behavior

Attachment of DU145 and PC3 cells was significantly inhibited when cells were treated with curcumin (0.2 µg/mL) or with curcumin plus visible light (5 min at 1.65 J/cm^2^). Curcumin application in concert with irradiation triggered maximum effects (Figure 6A, left). Irradiation alone did not cause any alteration in the binding behavior, compared to controls. Interaction of the tumor cells with an endothelial monolayer was also modified, verified by a loss of tumor-endothelial cell contacts in the presence of curcumin and light. A slight, transient decrease after 60 min was visible under curcumin application alone for DU145 cells (Figure 6A, right). To exclude toxic effects of curcumin plus light on HUVEC, necrosis and apoptosis have also been evaluated. Neither necrosis nor apoptosis increased under light, curcumin, or curcumin plus light, independent of whether HUVEC were treated for 30, 60, or 120 min (data not shown). The influence of curcumin on PC3 and DU145 cell adhesion may, therefore, not be related to HUVEC death.

Motile activity of DU145 and PC3 cells was significantly suppressed following curcumin–light application, but not when the cells were treated with either light or curcumin alone (Figure 6B). The adhesion receptor CD44 was also downregulated by the curcumin plus light combination but not when the tumor cells were exposed to curcumin or light alone (Figure 6C).

### 2.7. Integrin Expression

Figure 7 shows the integrin expression profile on DU145 and PC3 cells. The integrin subtypes α2, α3, α5, α6, αV, and β1 were all strongly present on the DU145 and PC3 cell surface. Integrin α1 was only marginally detectable in both tumor cell lines. Slight fluorescence signals of DU145 cells were recorded in the subtypes α4 and β3, whereas these integrins were not detected at all on PC3 cells. The β4 integrin was seen on DU145 and PC3 with PC3 > DU145 cells.

Curcumin (0.2 µg/mL) combined with visible light irradiation (5 min at 1.65 J/cm^2^) significantly altered the expression level of some integrin subtypes. Curcumin and light exposure alone did not lead to any changes of the receptor expression profiles. The subtypes α5, α6, and αV were downregulated in DU145 and PC3 cells. Integrin β3 was reduced on DU145 but not on PC3, whereas the opposite was seen with β4—reduced on PC3 but not on DU145 (Figure 8A). Furthermore, the integrin subtypes α2 and α3 were diminished on PC3 but not on DU145 cells in the presence of curcumin and subsequent light irradiation.

Analysis of the integrin protein level in DU145 cells revealed a loss of the integrin α subtypes α2, α3, α5, and αV but an increased level of α6 with curcumin and light (Figure 8B and Appendix A). The integrin β subtype β1 was lost as well but β4 was not altered under the curcumin–light combination. Evaluation of integrin related signaling pointed to diminished expression of ILK but enhanced expression of FAK (both total and phosphorylated). In PC3 cells, the α-integrins α3, α5, and αV were diminished. However, the integrin α2 expression was similar between treated and non-treated cells. Integrin α6 was only slightly down-regulated by curcumin–light, compared to the untreated control. Integrin signaling was strongly influenced, and all ILK, FAK, and pFAK were lost when PC3 cells were exposed to curcumin–light.

## 3. Discussion

Several strategies have been proposed to overcome the poor bioavailability of curcumin seen in clinical studies. One of them has been to combine light irradiation with curcumin application. The present study demonstrates that visible light irradiation of prostate cancer cells treated with curcumin strongly enhances curcumin’s anti-tumor effects. Indeed, applying 0.1–0.4 µg/mL curcumin alone did not induce any changes in DU145 and PC3 cell growth. Earlier studies point to 8 µM (2.95 µg/mL) curcumin being necessary to suppress the growth of gastric cancer cell lines by 50% [17]. Even higher concentrations were required to block the growth of breast [18], neuroblastoma [19], or bladder cancer cells [20] with IC50 values of up to 80 µM (29.52 µg/mL) curcumin. Based on the MTT analysis employed in the present investigation, 0.1 µg/mL curcumin already significantly reduced DU145 and PC3 cell number, and 0.4 µg/mL curcumin completely stopped DU145 growth, when applied together with light. The same protocol led to substantial down-regulation of proliferative activity and destruction of tumor clones. Curcumin alone has been reported to half-maximally suppress DU145 and PC3 colony formation only at concentrations above 15–20 µM (5.54–7.38 µg/mL) [21], 100-fold higher than the concentration employed in the present investigation. Light exposure, therefore, greatly increases curcumin’s inhibitory effects on growth and proliferation in prostate cancer cells. The photodynamic influence on curcumin described here is not restricted to prostate cancer. Light also improves curcumin’s bioavailability in skin keratinocytes [22] and renal [23] and bladder cancer cells [24], pointing to a mechanism common to different cancer types.

Light plus curcumin application triggered a G2/M cell cycle arrest in both cell lines evaluated. A- and B-type cyclins, along with their corresponding partners, CDK2 and CDK1, are crucial for mitotic transit with reduced expression correlated to G2/M-phase arrest [24,25]. A similar scenario might hold true in the present model, since cyclins, along with CDKs, were equally well suppressed in PC3 and DU145 (being aware that pCDK2 was not detectable in DU145). In contrast to curcumin’s effect on G2/M in both cell lines, diverse activity was exerted on the S- and G0/G1-phase. In DU145, G0/G1 phase cells were diminished, whereas a reduced number of S-phase cells was seen with PC3 cells. Curcumin, therefore, does not consistently influence mitotic progression in both tumor subtypes since differences in cell cycle modulation are apparent. In different bladder cancer cell lines, curcumin has also been shown to block growth with either an accumulation in the G2/M or in G0/G1 phases, depending on the tumor subline [24]. Western blot analysis revealed a different expression profile of the mTOR members, Rictor and Raptor, in the presence of curcumin, dependent on the cell line. Phosphorylated Rictor and Raptor were completely lost in DU145 but only slightly diminished in PC3, which indicates a decisive role of mTOR-signaling in the fine tuning of cell cycle phase transition. Whether excessive suppression of Rictor and Raptor indeed contributes to lowering DU145 G0/G1-phase cells is unclear since the relevance of pRictor and pRaptor depletion for cell cycle regulation has not yet been evaluated. It should also be considered that the curcumin–light combination increased the amount of apoptotic tumor cells, whereby apoptosis induction was more pronounced in DU145 than in PC3 cells. Curcumin (20 µM; 7.38 µg/mL) has been shown to increase G2/M, decrease G1 phase 22rv1 cells, and concomitantly reduce the anti-apoptotic protein Bcl-xL [26]. Furthermore, caspase-related apoptosis was exerted on DU145 and PC3 cells when exposed to 50 µM (18.45 µg/mL) curcumin [27]. Thus, the modification of apoptotic-related intracellular signaling by curcumin and light may also be coupled to cell cycle halt.

The reduced expression of Akt and pAkt by low-dosed curcumin is notable. Akt serves as a major target to block prostate cancer growth, proliferation, adhesion, and invasion. A phase 2 clinical trial has shown that blocking Akt, compared to second-line chemotherapy [28], is associated with tumor shrinkage and delayed disease progression in the significant subset of patients with metastatic castration-resistant prostate cancer. Randomized trials combining an Akt inhibitor with androgen deprivation (abiraterone, enzalutamide) are currently underway [29]. Overall, light irradiation of prostate cancer cells is shown in the present investigation to profoundly elevate curcumin’s bioavailability, with a concentration of 0.2 µg/mL distinctly blocking tumor proliferation by interacting with the CDK-cyclin axis and the Akt-mTOR pathway.

Apart from being a growth suppressor, low-dosed curcumin combined with light irradiation also acted on adhesion and chemotactic properties of the tumor cells. One-hundred-fold higher concentrations are required to evoke similar responses by curcumin alone [30,31]. Since both adhesion and migration were diminished equally well by combined low-dosed curcumin and light, reduced migration could either be a consequence of the diminished attachment rate or adhesion and migration could have been regulated by separate mechanisms. Fluorescence analysis showed particular integrin subtypes to be altered by curcumin and light. Expression of α5, αV, and α6 was downregulated on both DU145 and PC3 cells. Intriguingly, integrin α6 promotes survival and serves as a direct transcriptional target of the androgen receptor once the tumor fails to respond to androgen deprivation. α6 also seems to be involved in resistance development towards PI3K/Akt/mTor inhibitors [32]. The analysis of tumor specimens derived from prostate cancer patients has provided evidence that α5 is absent in the primary tumor but highly expressed in bone metastases, supporting a role of this subtype in bone colonization [33]. Integrin αV was found to be strongly present in high-grade prostatic adenocarcinomas and furthermore co-expressed with α5 in osseous metastases [33]. Based on these findings, α5, αV, and α6 may constitute important target candidates to slow bone dissemination of aggressive prostate cancer. Several antibodies with inhibitory activity on specific integrin subtypes have meanwhile been developed and are under investigation [34]. Still, to prevent the outcome of adaptive resistance, which may occur under single integrin targeting, combined targeting of several integrin members has been recommended for a future therapeutic regimen [35]. Curcumin could be the ideal agent, due to its suppressive action on the integrins α5, αV, and α6.

Aside from α6, α5 and αV, further integrin family members were removed from the cell surface by curcumin and light. However, this did not occur consistently. Different responses of DU145 and PC3 cells were observed with β3 expression (down-regulated on DU145) and the integrins α2, α3, β1, and β4 (down-regulated on PC3). Earlier studies support the hypothesis that differences in the molecular fine tuning of invasive processes may (at least in part) depend on the initial integrin expression pattern [36]. The integrin subtype β3 was detected on DU145 but not on PC3 cells, and distinctly more β4 receptors were detected on PC3 compared to DU145. Regardless of the initial integrin content, potent invasion-preventive properties were induced in vitro by curcumin coupled with light irradiation. This was triggered by the loss of specific but not identical integrin subtypes on the surface membranes of the PC3 and DU145 cells. Protein levels of α5 and αV were also diminished, whereas the α6 protein was elevated by curcumin plus light (DU145). Presumably, α3 and α5 were detached from the tumor cells, whereas α6 (DU145) was translocated into the cytoplasm. Both α3 and β1 were also diminished, although cell surface expression was only downregulated on PC3 but not on DU145 cells. Different mechanisms could be assumed, with the dissociation of α3 and β1 from the PC3 membrane, but intracellular degradation of α3 and β1 in DU145. The response of FAK is difficult to interpret. Although the integrin-related signaling protein ILK was reduced by curcumin and light in both cell lines, FAK/pFAK was lost in PC3 but elevated in DU145 cells. More integrin subtypes have been modulated in PC3 cells by curcumin plus light, which, speculatively, may create a stronger suppression of integrin-related signaling. Still, this does not explain why FAK/pFAK increased in DU145 cells. Since the integrin α6 protein was elevated in DU145 but down-regulated in PC3 cells, a FAK-α6-cross-communication should be considered. No investigations to date have dealt with this issue.

A low concentration of curcumin has been shown to potently reduce the proliferative and invasive activity of androgen-independent prostate cancer cells by interfering with the CDK-cyclin, the Akt-mTOR pathway, and integrin-related signaling. These effects were not apparent unless the tumor cells had been exposed to visible light. Several techniques have been reported, whereby light has been successfully applied to tumor tissue in in vivo models and in clinical practice. Neuschmelting et al. introduced an optical fiber into renal cell tumors in mice to successfully activate the photosensitizer WST11 by laser illumination with a wavelength of 750 nm or multispectrally of 700–800 nm [37]. Kroeze et al. inserted a cylindrical fiber with an effective diffuser in the center of renal tumors using an intravenous catheter which was then illuminated with a wavelength of 652 nm [38]. It has also been suggested to expose the tumor bed to light after tumor resection and curcumin administration [39]. Novel data point to the vascular targeted photodynamic therapy with WST11 (TOOKAD) as a novel treatment option for localized prostate cancer [40,41]. Overcoming the poor bioavailability of curcumin with light application is, therefore, a viable step towards implementing the clinical use of ingested curcumin.

## 4. Materials and Methods

### 4.1. Cell Culture

Prostate carcinoma androgen-insensitive DU145 (mutations in CDKN2A, RB1, TP53) and PC3 (mutations in PTEN, TP53) cell lines were obtained from DSMZ (Braunschweig, Germany). The human castration-resistant tumor cell lines were grown and subcultured in RPMI 1640 medium supplemented with 10% fetal calf serum (FCS), 1% Glutamax (all Gibco/Invitrogen, Karlsruhe, Germany), 2% Hepes buffer and 1% penicillin/streptomycin (both Sigma-Aldrich, Taufkirchen, Germany), at 37 °C in a humidified incubator with 5% CO_2_. Subcultures from passages 5–30 were selected for experimental use.

### 4.2. Drug Treatment and Light Application

Curcumin was stored at −20 °C and diluted in cell culture medium to the final concentration of 0.1–1 µg/mL (0.271–2.71 µM). Control cell cultures were cultivated in cell culture medium without treatment. To exclude significant effects of low-dosed curcumin and light on its own, two additional control groups were assessed. Treatment was performed at least 24 h prior to the experiment in the desired concentration. Cells were treated for 1 h with curcumin and/or were irradiated with visible light for 5 min (1.65 J/cm^2^). During that time the culture medium was replaced by PBS (Sigma-Aldrich). The untreated control and the curcumin treated control group were stored in PBS and kept light-protected to ensure identical experimental conditions. Thereafter, the cells were cultured in cell culture medium until assays were performed.

### 4.3. Measurement of Tumor Cell Growth and Proliferation

Cell growth was measured using the 3-(4,5-dimethylthiazol-2-yl)-2,5-diphenyltetrazolium bromide (MTT) dye reduction assay (Roche Diagnostics, Penzberg, Germany). Tumor cells (100 µL, 1 × 10^4^ cells/mL) were plated into 96-well tissue culture plates. After 24, 48, and 72 h, MTT (0.5 mg/mL) was added for an additional 4 h. The reaction was stopped by lysing the cells in a buffer containing 10% SDS in 0.01 M HCl. After incubating the plates at 37 °C, 5% CO_2_ overnight, the absorbance at 570 nm was measured for each well using a microplate proliferation enzyme-linked immunosorbent assay (ELISA) reader. Each experiment was performed in triplicate. After subtracting background absorbance, results were expressed as mean cell number.

Cell proliferation was measured using a BrdU cell proliferation enzyme-linked immunosorbent assay (ELISA) kit (Calbiochem/Merck Biosciences, Darmstadt, Germany). Tumor cells (50 μL, 1 × 10^5^ cells/mL), seeded onto 96-well plates, were incubated with 20 μL BrdU-labeling solution per well for 8 h, fixed and detected using anti-BrdU mAb according to the manufacturer’s instructions. Absorbance was measured at 450 nm using a microplate ELISA reader.

### 4.4. Apoptosis

The influence of apoptosis with regard to tumor cell growth was assessed with the Annexin V-FITC Apoptosis Detection kit (BD Pharmingen, Heidelberg, Germany) that quantifies binding of Annexin V/propidium iodide. Tumor cells, incubated with SFN (controls were without SFN) were washed twice with PBS, and incubated with 5 μL Annexin V-FITC and 5 μL propidium iodide in the dark for 15 min at room temperature. Cell numbers were determined by flow cytometry (FACScalibur, BD Biosciences, Heidelberg, Germany). Percentages of early and late apoptotic, necrotic, and vital cells in each quadrant were calculated using CellQuest software (BD Biosciences). Apoptosis was verified by the evaluation of the PARP protein expression level (clone 46D11; Cell Signaling, Danvers, MA, USA). β-actin (1:1000; clone AC-15; Sigma-Aldrich, Taufenkirchen, Germany) served as the internal control.

### 4.5. Clonogenic Growth

Five-hundred single PC3 or DU145 cells (treated with 0.2 µg/mL curcumin and irradiated versus non-irradiated or non-treated) were transferred to 6-well plates. Following 5 to 10 days incubation without medium change, cell colonies were fixed and counted. Clones of at least 50 cells were counted as one colony.

### 4.6. Cell Cycle Analysis

Cell cycle analysis was carried out with subconfluent tumor cells. Tumor cell populations were stained with propidium iodide, using a Cycle TEST PLUS DNA Reagent Kit (BD Pharmingen), and then subjected to flow cytometry with a FACScan flow cytometer (Becton Dickinson) after 24 h of cultivation. A total of 10,000 events were collected for each sample. Data acquisition was carried out using Cell-Quest software and cell cycle distribution was calculated using the ModFit software (BD Biosciences). The number of gated cells in the G1, G2/M, or S-phase is presented as a percentage of the total cell number.

### 4.7. Western Blot Analysis

To investigate the level of the cell cycle and movement related proteins in the three cell lines, tumor cell lysates were applied to 7% to 12% polyacrylamide gel and electrophoresed for 90 min at 100 V. The protein was then transferred to nitrocellulose membranes (1 h, 100 V). After blocking with nonfat dry milk for 1 h, the membranes were incubated overnight with monoclonal antibodies directed against the cell cycle proteins: CDK1/Cdc2 (IgG1, clone 1), pCDK1/Cdc2 (IgG1, clone 44/CDK1/Cdc2 (pY15)), CDK2 (IgG2a, clone 55), Cyclin A (IgG1, clone 25), Cyclin B (IgG1, clone 18; all: BD Pharmingen), pCDK2 (Thr160; Cell Signaling). The mechanistic target of rapamycin (mTOR) pathway was investigated by using the following monoclonal antibodies: Raptor (24C12), phospho Raptor (Ser792), Rictor (D16H9), phospho Rictor (Thr1135, D30A3, all Cell Signaling), PKBα/Akt (IgG1 clone 55) and anti-phospho Akt (pAkt; IgG1, Ser472/Ser473, clone 104A282; both: BD Pharmingen).

Integrins, adhesion or chemotaxis-related proteins were incubated as described before: FAK (IgG1, clone 77), pFAK (IgG1, pY397; clone 18) ILK (IgG1, clone 3), Integrin alpha 5 (clone 1), Integrin alpha V (clone 21), Integrin beta 1 (clone 18) and Integrin beta 4 (clone 7; all BD Pharmingen), Integrin alpha 2 (clone P1E6) and Integrin alpha 3 (both Millipore, Damrstadt, Germany), Integrin alpha 6 (Cell Signaling).

HRP-conjugated goat anti-mouse IgG and HRP-conjugated goat anti-rabbit IgG (both: 1:5000; Upstate Biotechnology, Lake Placid, NY, USA) served as the secondary antibody. The membranes were briefly incubated with ECL detection reagent (ECL; Amersham/GE Healthcare, München, Germany) to visualize the proteins and then analyzed by the Fusion FX7 system (Peqlab, Erlangen, Germany). β-actin (1:1000; clone AC-15; Sigma-Aldrich) served as the internal control.

### 4.8. Adhesion to Immobilized Collagen

Six-well plates were coated with collagen G (extracted from calfskin, consisting of 90% collagen type I and 10% collagen type III; Biochrom, Berlin, Germany; diluted to 400 µg/mL in PBS) overnight. Plastic dishes served as the background control. Plates were washed with 1% BSA (bovine serum albumin) in PBS to block nonspecific cell adhesion. 0.5 × 10^6^ tumor cells were then added to each well and left for 60 min incubation. Subsequently, non-adherent tumor cells were washed off, and the remaining adherent cells were fixed with 1% glutaraldehyde and counted microscopically. The mean cellular adhesion rate was calculated from five different observation fields (5 × 0.25 mm^2^).

### 4.9. Adhesion to Endothelial Cells

Twenty-four hours before the start of the experiment, HUVECs were seeded onto 6-well plates (Sarstedt, Nürnbrecht, Germany) and filled with 2 mL HUVEC medium (M199) so that a subconfluent cell lawn was present on the day of the experiment. Three plates of four wells each were prepared for each cell line. Before the experiment, half of the HUVEC medium was removed, and 0.5 × 10^6^ cells/mL added. One well each was filled with cells from one of the different treatments (control, light exposure without curcumin, curcumin without light exposure, and curcumin with light exposure). Plates were incubated for 30 min, one hour and two hours at 37 °C and 5% CO_2_. Subsequently, the cells were fixed with 1% glutaraldehyde. Adherent tumor cells were counted in five different fields of a defined size (5 × 0.25 mm^2^) using a phase contrast microscope and the mean cellular adhesion rate was calculated.

### 4.10. Chemotaxis

Serum-induced chemotactic movement was examined using 6-well Transwell chambers (Greiner, Frickenhausen, Germany) with 8 µm pores. 0.5 × 10^6^ DU145 or PC3 cells/mL of the respective treatment were placed in the upper chamber in serum-free medium. Serum free medium in the upper chamber and 10% serum in the lower chamber provided the serum gradient necessary for tumor cell migration in this model. After 20 h incubation, fixation (1% glutaraldehyde) and staining (hematoxylin), the upper surface of the Transwell membrane was gently wiped with a cotton swab to remove cells that had not migrated. Cells that had moved towards the serum gradient were counted microscopically. The mean migration rate was calculated from five different observation fields (5 × 0.25 mm^2^). In addition, the metastasis marker CD44 (Cell Signaling) was evaluated by Western blotting using β-actin as the internal control.

### 4.11. Integrin Expression

Tumor cells were washed in blocking solution (PBS, 0.5% BSA) and then incubated for 60 min at 4 C with phycoerythrin (PE)-conjugated monoclonal antibodies directed against the following integrin subtypes: anti-α1 (IgG1; clone SR84), anti-α2 (IgG2a; clone 12F1–H6), anti-α3 (IgG1; clone C3II.1), anti-α4 (IgG1; clone 9F10), anti-α5 (IgG1; clone IIA1), anti-α6 (IgG2a; clone GoH3), anti-β1 (IgG1; clone MAR4), anti-β3 (IgG1; clone VI-PL2) and anti-β4 (IgG2b; clone 439–9B; all: BD Pharmingen, Heidelberg, Germany) or anti-αV (IgG1, clone 13C2, SouthernBiotech, Birmingham, AL, USA). Integrin expression of tumor cells was then measured using a FACscan (BD Biosciences, Heidelberg; FL-2H (log) channel histogram analysis; 1 × 10^4^ cells/scan) and expressed as mean fluorescence units. A mouse IgG1-PE (MOPC-21), IgG2a-PE (G155–178) or rat IgG2b-PE (R35-38; all: BD Biosciences) was used as an isotype control.

### 4.12. Statistics

All experiments were performed three to six times. Statistical significance was calculated with the Wilcoxon–Mann–Whitney U test or the *t*-test. Differences were considered statistically significant at a *p* value less than 0.05.

## 5. Conclusions

Low dosed curcumin coupled with visible light application led to reduced proliferation and growth of PC3 and DU145 cells in vitro and was associated with a cell-cycle arrest in the G2/M-phase. Cellular adhesion and invasion capacity were also reduced, presumably by modifying cell specific integrin subtypes. The integrative application of combined curcumin and visible light may enhance the anti-tumor potential of conventional treatment regimens.

## Figures and Tables

**Figure 1 ijms-22-09966-f001:**
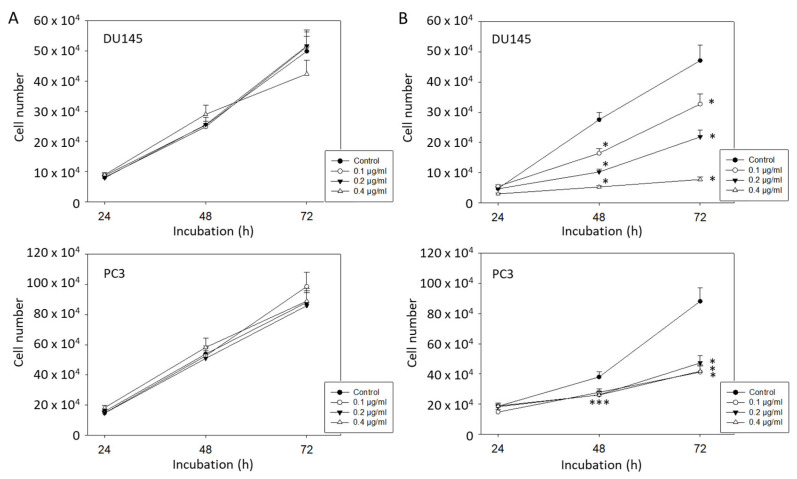
(**A**): Growth of DU145 and PC3 prostate cancer cells without irradiation. Cells were treated with 0.1, 0.2, or 0.4 µg/mL curcumin and then assessed by MTT assay. Controls remained untreated. (**B**): Growth of DU145 and PC3 prostate cancer cells treated with curcumin and visible light (5 min at 1.65 J/cm^2^). Controls were irradiated without curcumin. Each experiment was performed in triplicate and repeated 6 times. Data from one representative experiment is shown. Error bars indicate standard deviation; * indicates significant difference to controls. *** related to all 0.1, 0.2, and 0.4 µg/ml.

**Figure 2 ijms-22-09966-f002:**
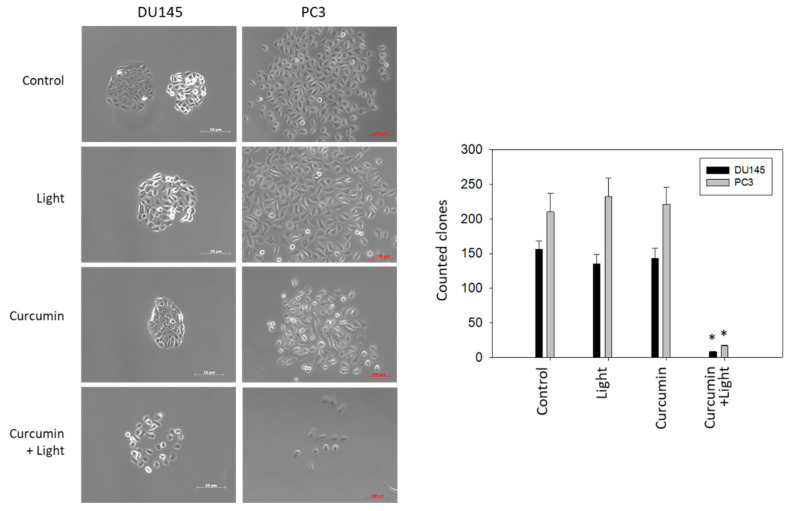
Left panel: Morphologic alterations of DU145 and PC3 clones after curcumin (0.2 µg/mL), visible light (5 min at 1.65 J/cm^2^) or curcumin, and light exposure. Controls remained untreated. Pictures were taken after 7 days of incubation. Right panel: Influence of visible light (5 min at 1.65 J/cm^2^), curcumin (0.2 µg/mL), or curcumin–light combination on clonogenic growth, compared to untreated controls. Error bars indicate standard deviation; * indicates significant difference to untreated controls. n = 3.

**Figure 3 ijms-22-09966-f003:**
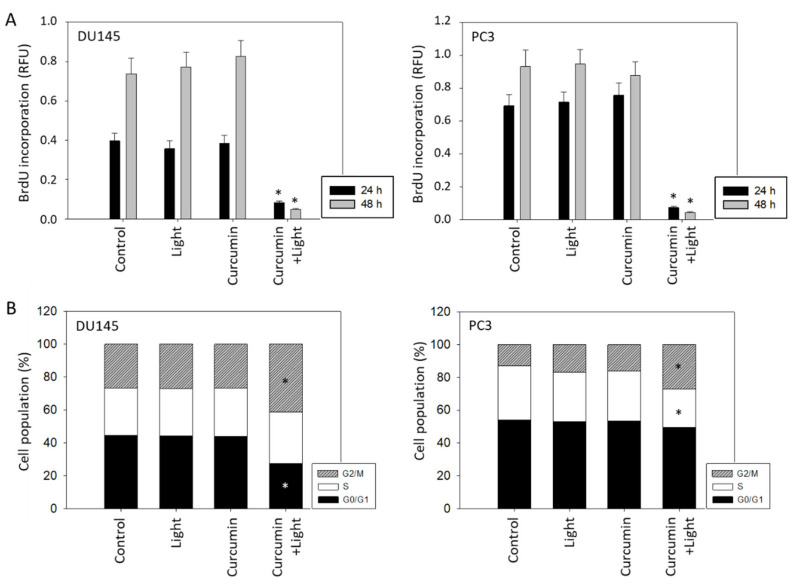
(**A**): BrdU incorporation under visible light exposure (5 min at 1.65 J/cm^2^), application of curcumin (0.2 µg/mL) or both curcumin and light irradiation. Controls remained untreated. Values are shown after 24 and 48 h incubation. Each experiment was done in triplicate and repeated 5 times. Data from one representative experiment is shown (n = 6). Error bars indicate standard deviation. (**B**): Influence of curcumin, light or curcumin plus light on proportionate G0/G1-, S-, and G2/M-phases of the cell cycle in DU145 and PC3 cells (24 h incubation) (n = 3; * indicates significant difference to untreated controls).

**Figure 4 ijms-22-09966-f004:**
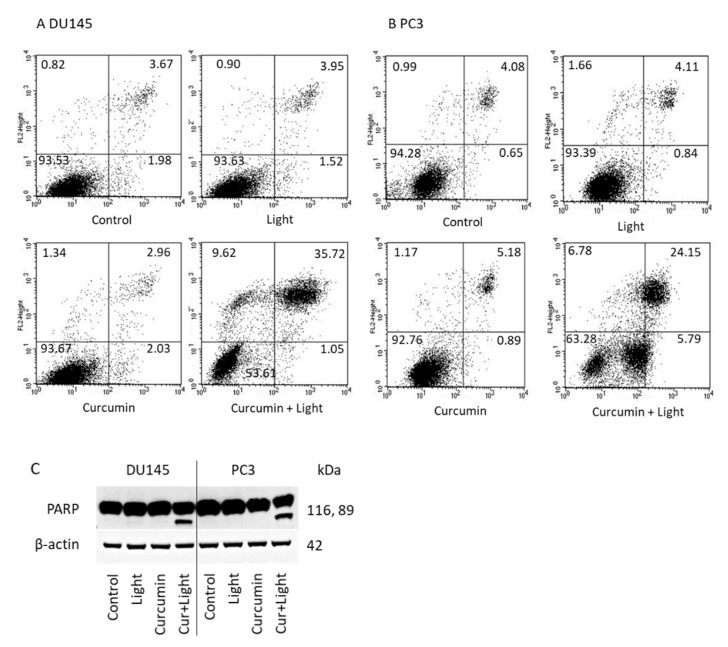
Induction of apoptotic events. DU145 (**A**) and PC3 (**B**) prostate cancer cells were incubated with visible light alone (5 min at 1.65 J/cm^2^) or 0.2 µg/mL curcumin alone for 24 h or were irradiated in the presence of curcumin (curcumin + light). Controls were incubated without curcumin or light. Early and late apoptosis was evaluated as indicated in methods. Dot blots of one representative test of three are shown. (**C**): PARP expression in DU145 and PC3 cells in the presence of visible light, curcumin, or curcumin + light (Cur + Light). Controls remained untreated. One representative result is shown.

**Figure 5 ijms-22-09966-f005:**
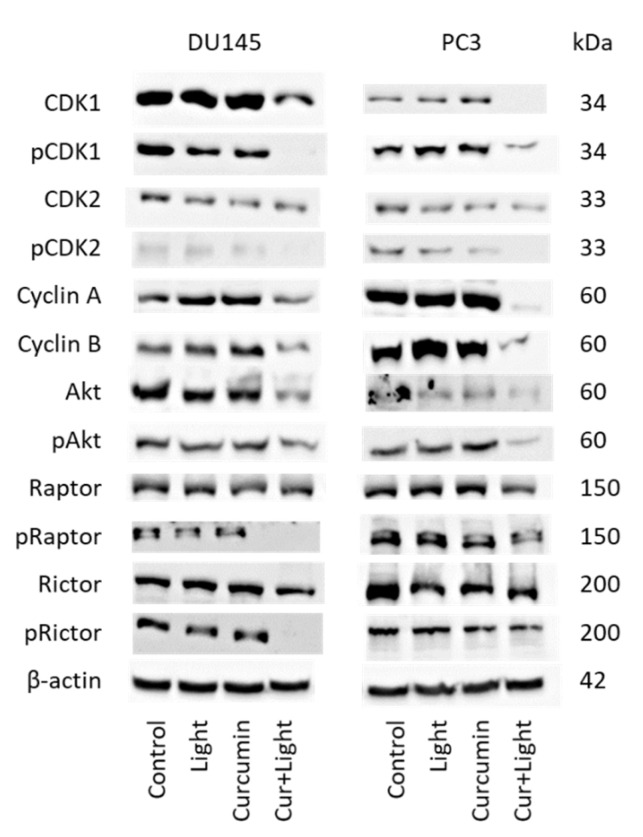
Protein expression profile of cell cycle regulating proteins in the presence of visible light (5 min at 1.65 J/cm^2^), curcumin (0.2 µg/mL) and the combination of curcumin plus light (Cur + Light) on DU145 and PC3 cells. Controls remained untreated. The protein isolation was carried out 24 h after treatment. ß-actin was used as the internal control. Each experiment was repeated 3 times. Data from one representative experiment is shown.

**Figure 6 ijms-22-09966-f006:**
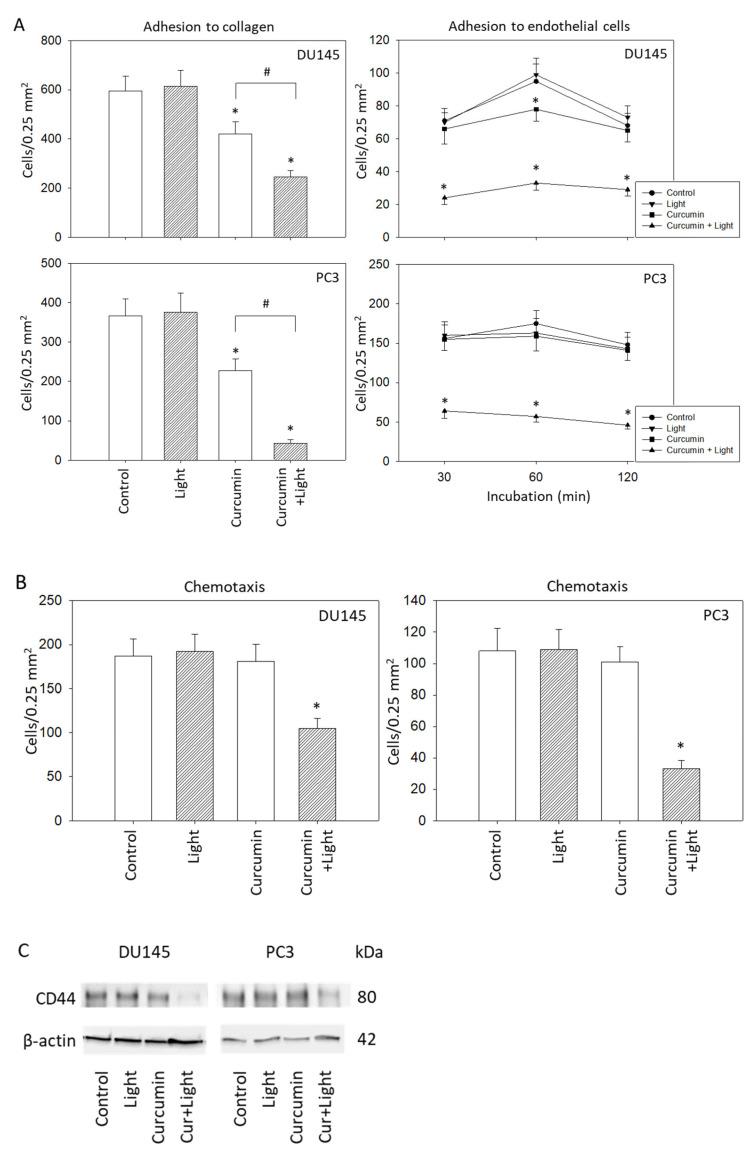
(**A**): Adhesion characteristics of DU145 and PC3 cells following exposure to visible light (5 min at 1.65 J/cm^2^), curcumin (0.2 µg/mL) or light alone, or to the curcumin–light combination. Controls depict untreated cells. The left panel of (**A**) shows cell binding to immobilized collagen, the right panel of (**A**) shows cell adhesion to endothelial cells. (**B**): Motility of treated versus non-treated DU145 and PC3 cells. Mean values were calculated from five counts; error bars indicate standard deviation; * indicates significant difference to the control; # indicates significant difference between the curcumin treated and the curcumin–light treated cells. (**C**): CD44 expression in DU145 and PC3 cells in the presence of visible light, curcumin, or curcumin+light (Cur + Light). Controls remained untreated. One representative result is shown.

**Figure 7 ijms-22-09966-f007:**
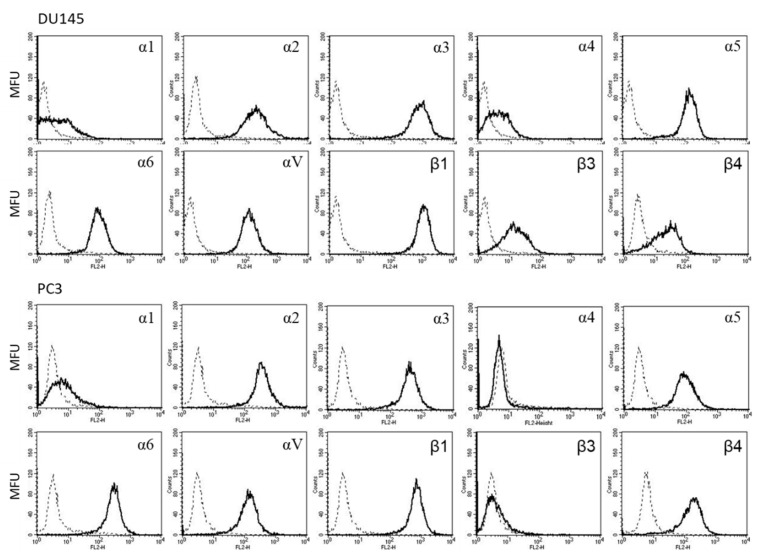
Surface expression of integrin α and β subtypes on DU145 and PC3 cells. Counts indicate mean fluorescence units (MFU). One representative of three separate experiments is shown. Solid line: specific fluorescence; dashed line: isotype control.

**Figure 8 ijms-22-09966-f008:**
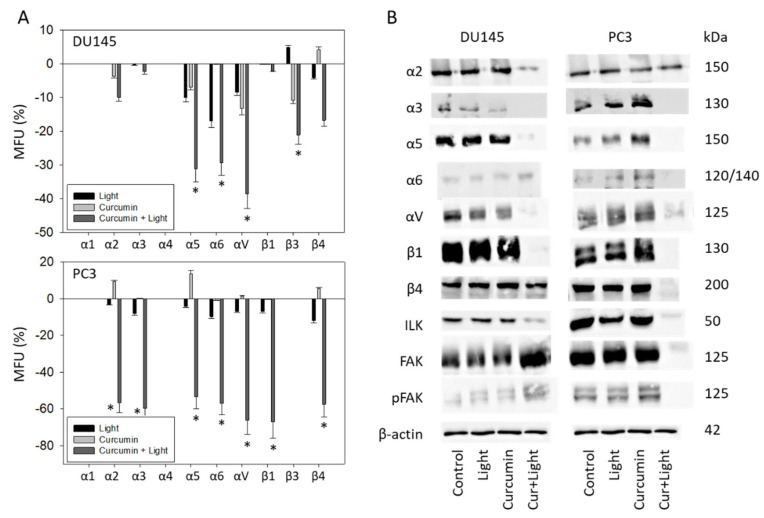
(**A**): Expression level of integrin α and β subtypes following visible light (5 min at 1.65 J/cm^2^), curcumin (0.2 µg/mL), or curcumin + light exposure. The upper panel depicts integrin expression for DU145 cells, the lower panel for PC3 cells. All values are related to untreated controls set to 100%. Means ± SD of n = 4; error bars indicate standard deviation; * indicates significant difference to corresponding control. (**B**): Integrin α and β, ILK and FAK protein analysis in DU145 and PC3 cells after exposure to light, curcumin, or curcumin + light combination (Cur + light). β-actin served as internal control. One representative of three separate experiments is shown. Controls remained untreated.

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
