# Peer review of "Growth, Proliferation and Metastasis of Prostate Cancer Cells Is Blocked by Low-Dose Curcumin in Combination with Light Irradiation"

_ijms, 2021, doi:10.3390/ijms22189966_

Round 1

Reviewer 1 Report

The manuscript ‘Growth, proliferation, and metastasis of prostate cancer cells is blocked by low-dose curcumin in combination with light irradiation through modulation of the Cyclin-CDK-axis and integrin α and β expression’ is interesting based on the combinatorial treatment. However, the minor changes might strengthen the manuscript.

Comments:

Minor:

  1. The title is too long; it would be nice to have a catchy title for the readers.
  2. Fig1, fig1A labels are inside the box, whereas for fig1B it's outside. Either of the labels needs to be changed. Consistent in all figures.
  3. Fig4, These data could be confirmed by another measure of apoptosis-like cleaved PARP expression or caspases.
  4. Fig6, These data could be confirmed by other metastatic marker expressions.
  5. Mutational status of DU145 and PC3.

Author Response

Answers to the comments of referee 1

Comment 1: The title is too long; it would be nice to have a catchy title for the readers.

Our answer: The title reads now: “Growth, proliferation and metastasis of prostate cancer cells is blocked by low-dose curcumin in combination with light irradiation”

Comment 2: Fig1, fig1A labels are inside the box, whereas for fig1B it's outside. Either of the labels needs to be changed. Consistent in all figures.

Our answer: Figure 1 has been modified accordingly.

Comment 3: Fig4, These data could be confirmed by another measure of apoptosis-like cleaved PARP expression or caspases.

Our answer: PARP expression has been investigated. The result is included in figure 4 (4C). Methods, chapter “Apoptosis”, reads now: “Apoptosis was verified by evaluation of the PARP protein expression level (clone 46D11; Cell Signaling, Danvers, USA). β-Actin (1:1.000; clone AC-15; Sigma-Aldrich, Taufenkirchen, Germany) served as the internal control. Results, chapter “Apoptosis”, reads: However, the percentage of apoptotic DU145 and PC3 cells considerably increased when treated with light and curcumin. PARP cleavage became evident in presence of light and curcumin but not in presence of light or curcumin alone (4C)”.

Comment 4: Fig6, These data could be confirmed by other metastatic marker expressions.

Our answer: We additionally evaluated the adhesion receptor CD44 by Western blotting. Results are depicted in figure 6 (6C). We have also added in Methods, chemotaxis: “In addition, the metastasis marker CD44 (Cell Signaling) was evaluated by Western blotting using β-actin as the internal control”. Results, Adhesion and migration behavior, reads: “The adhesion receptor CD44 was also down-regulated by the curcumin plus light combination but not when the tumor cells were exposed to curcumin or light alone (6C)”.

Comment 5: Mutational status of DU145 and PC3.

Our answer: Materials and Methods, chapter “Cell Culture”, reads now: “Prostate carcinoma androgen-insensitive DU145 (mutations in CDKN2A, RB1, TP53) and PC3 (mutations in PTEN, TP53) cell lines were obtained from …”.

Reviewer 2 Report

The manuscript entitled “Growth, proliferation and metastasis of prostate cancer cells is blocked by low-dose curcumin in combination with light irradiation through modulation of the Cyclin-CDK-axis and integrin α and β expression” investigated whether irradiation with visible light may enhance the inhibitory effects of low dosed curcumin on prostate cancer cell growth, proliferation and metastasis in vitro.

While the topic of this study is of interest in cancer research, there are a number of issues need to be addressed before reconsideration for publication. 

  1. The title is extensive: Please update the title.
  2. The introduction section is too short. It is not well defined if curcumin was already tested in prostate cancer cells lines or not. The authors should specify preliminary published data on the effect of curcumin in this type of tumor. From my point of view, there are already few studies showing the effect of curcumin in prostate cancer cells, thus it is not clear the novelty of this study.
  3. The authors used two metastatic cancer lines to evaluate the effect of this treatment. Could you please comment why you did not choose a primary cancer cell line and the rationality to only use metastatic cells?
  4. In the adhesion to endothelial cells did these cells contacted with curcumin and irradiation? Did you evaluate the effect of this treatment on endothelial cells viability?
  5. Xenograft mouse models of at least one of the cell lines are required to assess the tumor targeting ability of curcumin+irradiation.

Small considerations:

  1. In vitro should be written in italic
  2. The abstract should be presented without headings.

Author Response

Answers to the comments of referee 2

Comment 1: The title is extensive: Please update the title.

Our answer: The title reads now: “Growth, proliferation and metastasis of prostate cancer cells is blocked by low-dose curcumin in combination with light irradiation”.

Comment 2: The introduction section is too short. It is not well defined if curcumin was already tested in prostate cancer cells lines or not. The authors should specify preliminary published data on the effect of curcumin in this type of tumor. From my point of view, there are already few studies showing the effect of curcumin in prostate cancer cells, thus it is not clear the novelty of this study.

Our answer: The referee is absolutely correct. Several studies have already been published dealing with curcumin. We would like to emphasize in this context that we did not state out in the manuscript that we are the first one concentrating on this issue. We rather have discussed our results in the context of the literature and included the respective references (Ref. 6, 21, 26, 27, 30, 31). The novelty of our study concerns the curcumin-light combination protocol which is shown here to profoundly enhance curcumin’s bioavailability. We have now included further information into the introduction chapter which reads: “All of these approaches, however, have not led to satisfactory bioavailability. This is also true with respect to the influence of curcumin on prostate cancer. Although distinct effects have been noted in in vitro approaches, and although curcumin supplementation might have beneficial effects on some parameters related to prostate diseases [13], the overall clinical potential remains unsatisfactory. In fact, a recent interim analysis clearly showed that adding curcumin to metastatic castration-resistant prostate cancer patients was not effective in terms of progression-free survival [14].

Recently, Bernd presented evidence of increased antitumor properties of curcumin on human epidermal keratinocytes in vitro when irradiated with UVA or visible light [15]. The findings have been confirmed in a xenograft model, demonstrating tumor shrinkage when curcumin application was followed by irradiation [16]. The authors concluded that very low concentrations of curcumin might be sufficient to induce anti-tumor effects given that the treatment is combined with light irradiation. The present investigation deals with whether this strategy might be of value in treating prostate cancer”.

Comment 3: The authors used two metastatic cancer lines to evaluate the effect of this treatment. Could you please comment why you did not choose a primary cancer cell line and the rationality to only use metastatic cells?

Our answer: The metastatic and androgen-resistant tumor cells have been chosen, since non-responsiveness to the androgen deprivation therapy remains the prominent hurdle in cancer treatment. We agree that primary prostate cancer cells may distinctly enhance the value of our data. However, patient material can only be obtained from patients undergoing radical prostatectomy not from metastatic patients with acquired resistance. A further problem contains the low cell attachment rate, low growth activity and rapid necrosis of primary cell cultures. Therefore, the present study was initiated based on established cell lines.

Comment 4: In the adhesion to endothelial cells did these cells contacted with curcumin and irradiation? Did you evaluate the effect of this treatment on endothelial cells viability?

Our answer: In the context of the tumor cell adhesion assay, the influence of light, curcumin and curcumin+light has also been evaluated on HUVEC. Neither necrosis nor apoptosis increased under this regimen, independent on whether HUVEC were treated for 30, 60 or 120 min. The influence of curcumin on tumor cell adhesion may, therefore, not related to HUVEC death. We have now included in Results, chapter “Attachment and migration behavior”: “A slight, transient decrease after 60 min was visible under curcumin application alone for DU145 cells (figure 6A, right). To exclude toxic effects of curcumin plus light on HUVEC, necrosis and apoptosis have also been evaluated. Neither necrosis nor apoptosis increased under light, curcumin or curcumin plus light, independent on whether HUVEC were treated for 30, 60 or 120 min (data not shown). The influence of curcumin on PC3 and DU145 cell adhesion may, therefore, not be related to HUVEC death”.

Comment 5: Xenograft mouse models of at least one of the cell lines are required to assess the tumor targeting ability of curcumin+irradiation.

Our answer: Actually, a respective in vivo model (TOOKAD) has not been established yet. A PubMed search indicated three hits, whereby TOOKAD has only been evaluated in healthy dogs without treatment (Chevalier et al. J Urol. 2013;190:1946-53. J Urol. 2011;186:302-9. Huang et al. Photochem Photobiol Sci. 2007;6:1318-24). Due to still unsolved technical problems, we are yet not able to verify our in vitro data in an animal model. Therefore, we decided to deal with TOOKAD and further approaches in the discussion section (last para) to give an outlook how light irradiation can be carried out in the clinical setting. We hope that the referee will understand our problem and will be satisfied by the information given in “Discussion”. Independent on this, curcumin uptake into the cell cytoplasm has already been documented earlier (Rutz et al. Cancers. 2020;12:302). The paper is cited as reference 23.

Comment 6: In vitro should be written in italic. The abstract should be presented without headings.

Our answer: The comment has been taken care of.

Round 2

Reviewer 1 Report

The authors have addressed all the reviewer's comments. Hereby, the manuscript titled 'Growth, proliferation, and metastasis of prostate cancer cells is blocked by low-dose curcumin in combination with light irradiation' can be accepted for publication.

Reviewer 2 Report

Thank you very much for providing the revised version of your paper entitled “Growth, proliferation and metastasis of prostate cancer cells is blocked by low-dose curcumin in combination with light irradiation”.

With all the revisions I agree that this manuscript is ready for publication.